# The Implication of Low Dose Dimethyl Sulfoxide on Mitochondrial Function and Oxidative Damage in Cultured Cardiac and Cancer Cells

**DOI:** 10.3390/molecules26237305

**Published:** 2021-12-01

**Authors:** Nonhlakanipho F. Sangweni, Phiwayinkosi V. Dludla, Nireshni Chellan, Lawrence Mabasa, Jyoti R. Sharma, Rabia Johnson

**Affiliations:** 1Biomedical Research and Innovation Platform (BRIP), South African Medical Research Council, Tygerberg 7505, South Africa; phiwayinkosi.dludla@mrc.ac.za (P.V.D.); nireshni.chellan@mrc.ac.za (N.C.); Lawrence.Mabasa@mrc.ac.za (L.M.); jyoti.sharma@mrc.ac.za (J.R.S.); 2Centre for Cardio-Metabolic Research in Africa, Division of Medical Physiology, Faculty of Medicine and Health Sciences, Stellenbosch University, Tygerberg 7505, South Africa

**Keywords:** mitochondria, bioenergetics, oxidative stress, apoptosis

## Abstract

Although numerous studies have demonstrated the biological and multifaceted nature of dimethyl sulfoxide (DMSO) across different in vitro models, the direct effect of “non-toxic” low DMSO doses on cardiac and cancer cells has not been clearly explored. In the present study, H9c2 cardiomyoblasts and MCF-7 breast cancer cells were treated with varying concentrations of DMSO (0.001–3.7%) for 6 days. Here, DMSO doses < 0.5% enhanced the cardiomyoblasts respiratory control ratio and cellular viability relative to the control cells. However, 3.7% DMSO exposure enhanced the rate of apoptosis, which was driven by mitochondrial dysfunction and oxidative stress in the cardiomyoblasts. Additionally, in the cancer cells, DMSO (≥0.009) led to a reduction in the cell’s maximal respiratory capacity and ATP-linked respiration and turnover. As a result, the reduced bioenergetics accelerated ROS production whilst increasing early and late apoptosis in these cells. Surprisingly, 0.001% DMSO exposure led to a significant increase in the cancer cells proliferative activity. The latter, therefore, suggests that the use of DMSO, as a solvent or therapeutic compound, should be applied with caution in the cancer cells. Paradoxically, in the cardiomyoblasts, the application of DMSO (≤0.5%) demonstrated no cytotoxic or overt therapeutic benefits.

## 1. Introduction

Dimethyl sulfoxide (DMSO) is a versatile compound that is extensively used as a solvent in pharmacology and toxicology to enhance drug delivery, and dissolve numerous drugs and herbal extracts. However, in recent years, increasing evidence has demonstrated that the amphiphilic nature of DMSO allows it to influence diverse biological and medical processes, such as disease pathology and intervention [1,2]. Certainly, the exploration of this amphiphilic solvent in clinical research and cell biology continues to highlight avenues to be meticulously investigated to broaden its use in biomedical science. Generally, the applied concentrations of DMSO are often unreported due to its obvious and frequent use [3]. This, coupled with its apparent low toxicity at concentrations less than 10% and its classification as a class 3 solvent, which is the safest category with low toxic potential, has led to its ubiquitous use and widespread application [3,4,5]. The influence that DMSO has on cellular mechanisms has been implicated in the modifications of essential cellular structures, such as proteins and DNA, and has been thus, studied for its involvement in cancer and cardiovascular diseases [6,7].

Given the lack of long-term effective therapies against such complications, it remains imperative to study the pharmacological actions of compounds such as DMSO, particularly, by targeting the implications of its non-toxic low doses. Previously, a dose of 0.1% DMSO was shown to induce epigenetic modifications, which impaired the expression of genes involved in cellular senescence and DNA repair in a 3D maturing cardiac model [3]. In contrast, 1% DMSO exposure significantly improved the nuclear morphology and antioxidant status of dermal fibroblasts [8]. This improvement was determined to be even higher than that observed in known antioxidants, such as N-acetylcysteine [8] Similarly, in colon cancer cells, low DMSO doses (0.1–1.5%) were shown to reduce the cellular levels of reactive oxygen species (ROS) and the cells proliferative activity relative to the control [9]. Literature has further shown that the addition of DMSO (1%) to culture medium at the formation stage of cardiomyocyte progenitor cells stimulates the differentiation of pluripotent stem cells (PSCs) into cardiomyocytes by a 1.5-fold increase [10]. Conversely, the addition of DMSO before the initiation of embryoid bodies is reported to suppress the differentiation of PSCs and therefore, the formation of cardiomyocytes [10]. Engineered cardiac tissue, which is generated from differentiated PSCs, are reported to have enhanced engraftment rates, as well as increased survival and progressive maturation of human cardiomyocytes [11]. In cancer research, the differentiated PSCs are currently used to develop cancer-based vaccines and have reportedly inhibited the formation of new tumors in 75% adenocarcinomas [12]. While it is evident that the changes in cellular processes following DMSO exposure in the PSCs appear to have some beneficial properties, the direct effect of DMSO on the cardiomyocytes and cancer cells has not been clearly explored. As such, we investigated the biological effect of “non-toxic” low DMSO doses in an in vitro model of cardiac and cancer cells. In this study, special attention was paid into understanding the effects of DMSO on the most studied cytotoxicity parameters, namely, oxidative stress, mitochondrial dysfunction, and resultant apoptosis. 

## 2. Results

### 2.1. Cell Viability 

In the present study, a range of DMSO concentrations (0.001–3.7%) were selected to determine and compare their cytotoxic profile in the H9c2 cardiomyoblasts and MCF-7 breast cancer cells after a 6-day treatment period. The results obtained revealed a concentration-dependent effect of DMSO, with the highest concentration being cytotoxic to both cell lines relative to the control group (*p* < 0.001). However, contrary to previous reports, a concentration of 0.5% DMSO had no significant effects on the growth and survival of both cell lines. Interestingly, the H9c2 cells treated with concentrations less than 0.5% DMSO presented with significantly augmented cellular viability relative to the control group (*p* < 0.001 and *p* < 0.01) after 6 days of treatment (Figure 1A). These findings were comparable with the effects observed in the viability of the MCF-7 cells (Figure 1B).

#### 2.1.1. Mitochondrial Bioenergetics 

To determine the effect that DMSO has on the efficiency of mitochondrial oxidative phosphorylation and functionality, we quantified the parameters associated with native cellular respiration and respiration in the presence of known energy metabolism inhibitors. When comparing the energy phenotype inherent to the cardiomyoblasts and breast cancer cells, our data demonstrated that the MCF-7 cells were more energetic than the H9c2 cells, as they had a higher oxygen consumption rate (OCR) and glycolytic activity (Figure 2A). In contrast, the H9c2 cells had a more quiescent energy phenotype, with an inherently lower respiratory capacity than the MCF-7 cells (Figure 2A). In the case of DMSO, H9c2 cells treated with 3.7% DMSO presented with significantly impaired physiological mitochondrial OCR, as seen by the significant loss in the cell’s basal respiratory capacity (*p* < 0.001) when compared to the control (Figure 2B,F). This loss was concomitant with the observed reduction in the cells extracellular acidification rate (ECAR) (Figure 2D). Likewise, MCF-7 cells treated with 3.7% DMSO presented with decreased OCR, ECAR, and basal respiratory activity, relative to the untreated cells (Figure 2C,E,G). Interestingly, DMSO (≤0.5%) had no significant effect on the cardiomyoblasts ATP turnover and spare respiratory capacity when compared to the control group (Figure 2J,N). In contrast, DMSO at 3.7% reduced the MCF-7 cells ATP linked respiration, ATP turnover, and maximal respiration relative to the control (Figure 2I,K,M). 

Subsequently, we then quantified the respiratory flux ratios, which is an estimation of relative mitochondrial work and function. The results showed an increase in the H9c2 cells respiratory control ratio (RCR) following DMSO exposure with doses ≤ 0.5% (Table 1). We further observed a significant decrease in the cardiomyoblasts coupling efficiency (*p* < 0.001), relative to control, thus, indicating a significantly lower proportion of oxygen consumed to stimulate ATP production compared with that driving proton leak. Surprisingly, we observed no significant changes in the MCF-7 cells respiratory flux ratios following DMSO exposure with doses < 3.7% (Table 2). 

#### 2.1.2. Mitochondrial Membrane Potential

Mitochondrial damage is a prominent precursor of decreased cell viability. In the present study, H9c2 cells treated with 3.7% DMSO presented with a significant loss (*p* < 0.001) in cell viability (due to a loss in MMP) as seen in the JC-1 fluorescent images versus the control (Figure 3A). Similarly, chronic 3.7% DMSO exposure led to a significant loss (*p* < 0.001) in MMP and resultant mitochondrial deformities, as seen by the significant reduction in J-aggregate fluorescence intensity in the MCF-7 cells relative to the control group (Figure 3B). However, in both cell lines, the lower concentrations (0.001%, 0.009%, 0.067%, and 0.5%) had no effect on the structural and functional integrity of the mitochondria and were further found to be significantly less toxic than the higher 3.7% DMSO (*p* < 0.001) (Figure 3A,B).

#### 2.1.3. Oxidative Stress 

Impaired mitochondrial bioenergetics and the resultant loss in MMP can alter aerobic metabolism thus, stimulating increased ROS production. Here, the results demonstrated that intracellular ROS activity in the H9c2 cells exposed to 3.7% DMSO was significantly augmented when compared to the control group (*p* < 0.05) (Figure 4A). This increase in ROS was determined by the dramatic shift in DCFH-DA dye intensity, which is proportional to intracellular hydrogen peroxide concentrations. However, a steady reduction in ROS activity was observed when these cells were treated with DMSO doses ≤ 0.5% (Figure 4A). Interestingly, a significant increase, in a dose-dependent manner, in intracellular ROS activity was observed in MCF-7 cells treated with either 0.009% (*p* < 0.05), 0.067 (*p* < 0.05), 0.5% (*p* < 0.05), or 3.7% (*p* < 0.001) DMSO, relative to the control group (Figure 4B). Although not significant when compared to the control, a dose of 0.001% DMSO also elevated ROS in these cells (Figure 4B).

#### 2.1.4. Apoptosis 

The intricate role of DMSO-induced oxidative stress, which is often exacerbated by mitochondrial damage, accelerates cellular apoptosis. As such, the rate of cell death in H9c2 and MCF-7 cells was investigated following DMSO exposure. Here, only 3.7% DMSO treatment significantly increased early (lower right quadrant; *p* < 0.001) and late (upper right quadrant; *p* < 0.001) apoptosis, as could be seen by the significant reduction in the number of live (lower left quadrant) H9c2 cells (*p* < 0.001) relative to the control. The results further showed a significant increase in the number of necrotic (upper left quadrant) H9c2 cells when compared to the control (Figure 5A–E). In contrast, in the MCF-7 cells, DMSO exposure with doses ≤ 3.7% also led to a significant (*p* < 0.05) increase in early and late apoptosis, as demonstrated by enhanced annexin V positive cells (Figure 5F,H). The results further showed a significant increase in the number of cells going into late apoptosis in a dose dependent manner compared to the control (Figure 5I). While treatment with 3.7% DMSO led to a significant reduction in the number of live MCF-7 cells (*p* < 0.001), the results also demonstrated a significant increase in the rate of necrosis (*p* < 0.001) (Figure 5G,J).

## 3. Discussion

In many disease models, uncontrolled production of ROS remains a central mechanism accountable for excess accumulation of damaged organelles, translating to tissue injury and other related deleterious effects [13]. Uncontrolled ROS is a known source for the depletion of intracellular antioxidant defense systems, a process implicated in the generation of oxidative stress. Within the cardiovascular system, besides being the precursor of plaque formation through its attack on circulating lipid products to cause endothelial dysfunction [14], oxidative stress can directly cause tissue injury, leading to cardiac fibrosis and conditions such as cardiomyopathies [15]. In cancer research, although the role of oxidative stress is controversial [16], cancer cells display an abnormal redox homeostasis, with very high ROS shown to be cytotoxic [17]. Thus, exploring the role of ROS in redox signaling and in tumor propagation, development, and metastasis, including uncovering associated cellular mechanisms remains relevant to discover effective therapies. 

For example, while leading chemotherapeutic agents such as doxorubicin are effective against certain tumours, these drugs are known to present with undesirable side effects such as inducing cardiotoxicity when used for a prolonged period [18,19]. Certainly, current research [19,20], including that from our group [14], is targeting the use of antioxidant therapies to ameliorate oxidative stress-related detrimental effects consistent with the development of cancer and cardiac injury. Here, cultured cardiomyocytes and cancer cells were exposed to different doses of DMSO to understand their modulatory effect on oxidative stress-related parameters, such as mitochondrial function and apoptosis. This is especially important, since while low doses of DMSO can present with some antioxidant properties [21], this effect is known to vary with experimental models. Furthermore, as one of the commonly used solvents to deliver drugs or as a biological drug itself, it remains essential to broaden our understanding on the therapeutic properties of DMSO, especially its dose-dependent effect.

Both H9c2 and MCF-7 cells are widely used to investigate the therapeutic potential of drugs against oxidative stress mechanisms [14,22,23,24,25,26]. In the current study, these cells were exposed to DMSO doses, ranging from 0.001 to 3.7% for 6 days, before assessing endpoints such as cell viability, ROS production, efficiency of the mitochondrial respiratory capacity, changes in mitochondrial membrane potential, as well as markers of early and late apoptosis. Firstly, it was clear that DMSO doses ≤ 0.067% were not cytotoxic and, in fact, improved viability was observed for both cell lines. Alternatively, a dose of 0.5% had no effect on cell viability but, that of 3.7%, led to a significant reduction in cardiomyoblast viability. Conversely, both doses of 0.5 and 3.7% demonstrated signs of toxicity in the cancer cells. The next question was to determine the oxidative stress-related parameter that may explain improved cell viability with low DMSO doses (≤0.067%), or the cytotoxicity seen with high doses (>0.5%) in both cell lines. As one of the major organelles that can determine cellular fate, special attention was paid into understanding how the various doses of DMSO affected mitochondrial respiration. This aspect remained vital to explore since, as part of being the cell’s powerhouse, mitochondria are important for ATP production, controlling ROS production and antioxidant activity, as well as regulating apoptosis. The latter defines a highly ordered process that could be severely influenced by harmful stimuli, such as accelerated ROS activity and mitochondrial depolarization [27,28]. As a result, it has become necessary to collectively assess the therapeutic potential of any drug at the cellular level, by measuring its effect on cell viability, mitochondrial function, and apoptosis.

In the current study, we demonstrated that cardiomyoblasts have an inherently lower energy phenotype than breast cancer cells, which presented with a significantly higher oxygen consumption rate (OCR) and glycolytic profile. Here, the cardiomyoblasts displayed a physiologically lower respiratory capacity, which validated their quiescent energy phenotype. Upon DMSO exposure, with 0.001–0.067%, we observed a noticeable improvement in the cardiomyoblasts respiratory control ratio and ATP turnover, but not the breast cancer cells (Figure 3, Table 1 and Table 2). Instead, although not significant, these doses appeared to reduce the cancer cells mitochondrial bioenergetics. Briefly, the respiratory control ratio, represented the maximum factorial increase in mitochondrial OCR that could be attained above the leak oxygen prerequisite when driving the conversion of ADP into ATP [29]. This clearly explained the potential capacity of low DMSO doses to improve cell viability and maintain mitochondrial function, whilst protecting against apoptosis in the cardiomyoblasts; an outcome that was not observed in the cancer cells. While these experimental results still need to be confirmed in vivo, it appears that low DMSO doses (≤0.5%) may have had a positive effect on the cardiomyoblasts when considering processes that prevent oxidative damage, via improving mitochondrial respiration. Likewise, the findings of this study further demonstrated that the rate of apoptosis in the cardiomyoblasts treated with the “non-toxic” low DMSO doses (≤0.5%) was comparable to the severity of apoptosis observed in the untreated control group. These findings were supported by the preserved live population and reduced number of early and late apoptotic cardiomyoblasts. In contrast, the data showed that 3.7% DMSO exposure presented with severe cardiomyoblast toxicity, as shown by the increased number of apoptotic and necrotic cells.

Interestingly, in the breast cancer cells, 3.7% DMSO exposure impaired the cells oxidative phosphorylative capacity, which was demonstrated by the cells reduced basal respiration, ATP-linked respiration, ATP turnover, and maximal respiration. The data further highlighted that DMSO doses ≤ 3.7% triggers oxidative stress, in a dose dependent manner, as determined by the significant increase in ROS activity in the cancer cells. Further justifying its potential exploration as a possible therapeutic compound, was the significant increase in the number of early and late apoptotic cancer cells following treatment with DMSO (≤3.7%). Additionally, a much higher degree of necrosis was observed in these cells.

## 4. Materials and Methods

### 4.1. Reagents

Dimethyl sulfoxide (DMSO), Dulbecco’s phosphate-buffered saline (DPBS), Dulbecco’s Modified Eagle Medium (DMEM), tissue culture grade water, trypsin, and Hanks balanced salt solution (HBSS) were obtained from Lonza (Walkersville, MD, USA). Fluorescent probes, 5,5′, 6,6′-tetrachloro-1,1′, 3,3-tetraethylbenzimidazolyl-carbocyanine iodide (JC-1), and propidium iodide (PI) were obtained from Sigma-Aldrich (St Louis, MO, USA). Fetal Bovine Serum (FBS) was purchased from Thermo Fisher Scientific (Waltham, MA, USA). Annexin V and fluorescein conjugate (FITC annexin V) was purchased from Invitrogen (Carlsbad, CA, USA). The oxiselect ™ intracellular ROS Assay Kit (Green Fluorescence) was purchased from Cell Biolabs (San Diego, CA, USA). The XF Cell Mito Stress Kit was purchased from Agilent technologies (Santa Clara, CA, USA).

### 4.2. In Vitro Models

MCF-7 human breast cancer cells and H9c2 cardiomyoblasts were purchased from the American Type Culture Collection (ATCC, catalogue number HTB-22 and CRL-1446, respectively). Both cell lines were cultured in DMEM supplemented with 10% Fetal Bovine Serum (FBS, Thermo Fisher Scientific, Waltham, MA, USA) under standard tissue culture (TC) conditions (37 °C, 95% humidified air, and 5% CO_2_). The cytotoxic threshold of DMSO was investigated on the MCF-7 and H9c2 cells, using a series of concentrations (0.001, 0.009, 0.067, 0.5, and 3.7%). These concentrations were randomly selected from a previously conducted preliminary study on healthy cardiomyoblasts (Appendix A). Briefly, treatment was prepared in DMEM without phenol (Lonza, Walkersville, MD, USA), supplemented with 2% FBS. Prior to the experiments, the DMSO treatment was filter-sterilized using 0.22 μM syringe filter systems in a Class II Type A2 Biological Safety Cabinet. To assess the chronic effect of DMSO on the H9c2 and MCF-7 cells, the cells were treated every second day for 6 days, as demonstrated in Figure 6. Biochemical assays were conducted after terminating treatment.

### 4.3. Determination of Cell Viability

The effect of DMSO on mitochondrial activity, as a measure of cell viability, was assessed using the MTT (3-(4,5-dimethylthiazol-2-yl)-2,5-diphenyltetrazolium bromide) assay following an in-house protocol previously described [25]. Concisely, H9c2 and MCF-7 cells were seeded (0.8 × 10^5^/well) in 96-well clear plates and then treated as described above. Treatment was terminated by washing the cells with 100 µL DPBS. Subsequently, the cells were exposed to 100 µL MTT solution (2 mg/mL) and then incubated for 1 h under standard TC conditions. Cell viability was quantified by measuring absorbance on the SpectraMax^®^ i3x Multi-Mode Microplate Reader, at a wavelength of 570 nm.

### 4.4. Assessment of Mitochondrial Respiratory Capacity

Respiratory parameters associated with mitochondrial bioenergetics were measured in intact breast cancer cells and cardiomyoblasts using the XF Cell Mito Stress Kit (Seahorse Bioscience, Billerica, MA, USA) according to the manufacturer’s instruction, which is based on the sequential injection of oligomycin, carbonyl cyanide-4-(trifluoromethoxy) phenylhydrazone, rotenone, and antimycin, as per the manufacturer’s instructions. Seeding densities for the H9c2 (1 × 10^4^/well) and MCF-7 (5 × 10^3^/well) cells were determined from a previous study [25]. Concisely, cells were seeded into XF96-well cell culture microplates (Seahorse Bioscience, Billerica, MA, USA) after which, treatment was initiated after 48 h as described above. In preparation of the assay, the cells were incubated for 1 h with the Seahorse base assay medium (supplemented with 2 mM glutamine, 10 mM glucose, and 1 mM pyruvate). Hereafter, mitochondrial oxygen consumption rate (OCR) and extracellular acidification rate (ECAR) were quantified on the intact live cells using the Seahorse XF96 extracellular flux analyser (Seahorse Bioscience, Billerica, MA, USA). The data were normalized to protein concentrations using the Bio-Rad DC Protein assay (Bio-Rad, Hercules, CA, USA), as per the manufacturer’s instructions with the resultant absorbance read at 695 nm using the SpectraMax^®^ i3x Multi-Mode Microplate Reader. The results were expressed as pmol/min/mg protein for the OCR, and mpH/min/mg protein for the ECAR.

### 4.5. Assessing Changes in Mitochondrial Membrane Potential (ΔΨm)

The effect of DMSO on the mitochondrial integrity of the H9c2 and MCF-7 cells was assessed using the fluorescent JC-1 Assay Kit (Sigma-Aldrich, St Louis, MO, USA), as per the manufacturer’s instructions. Briefly, cells seeded (0.8 × 10^5^/well) in black clear bottom 96-well plates were stained with 100 µL JC-1 dye (8 µM) and then incubated for 45 min under standard TC conditions. Mitochondrial membrane potential (MMP) was then quantified by measuring the fluorescence intensity of J-aggregates at 590 nm and JC-1 monomers (at 529 nm using the SpectraMax^®^ i3x Multi-Mode Microplate Reader.

### 4.6. Quantification of Reactive Oxygen Species (ROS)

The OxiSelect^TM^ Intracellular ROS assay kit (Cell Biolabs, San Diego, CA, USA) was used to quantify DMSO-stimulated ROS production. Cells were seeded in 24-well plates (1 × 10^5^ cells/well) and treated as described above. Once treatment with DMSO was terminated, cells were washed with pre-warmed DPBS before being stained with 100 µL/well 2′,7′dichlorofluorescin diacetate (DCFH-DA, 20 µM) dye and then incubated for 30 min. Following incubation, the DCFH-DA dye was aspirated, and the cells were washed with pre-warmed DPBS. Thereafter, cells were trypsinized (150 µL) for either 6 min for the H9c2 cells, or 5 min for the MCF-7 cells, in an incubator under standard TC conditions. Trypsinization was deactivated by the addition of 300 µL pre-warmed media (DMEM, supplemented with 10% FBS). The cell suspension was then collected into 2 mL Eppendorf tubes and centrifuged for 5 min at 1500 rpm for H9c2 cells, and 120 RCF for the MCF-7 cells. Hereafter, the cells were resuspended in 150 µL DPBS and then placed on ice. ROS activity was quantified using the BD Accuri C6 flow cytometer (BD Biosciences, Franklin Lakes, NJ, USA).

### 4.7. Apoptosis Assay

DMSO-induced apoptosis was quantified by staining cells with Annexin V-FITC (Invitrogen, Carlsbad, CA, USA) and propidium iodide (PI, Sigma-Aldrich, St. Louis, MO, USA) as previously described [25]. Briefly, H9c2 and MCF-7 cells were seeded in 24-well plates (1 × 10^5^ cells/well) and treated as described above. The cells were trypsinized and collected into 2 mL Eppendorf tubes as previously reported [25]. The cells were then co-stained with 1.5 µL Annexin V and 1 µL PI (2 μg/mL) before being incubated in the dark for at least 10 min for the H9c2 cells, or 20 min for the MCF-7 cells. The rate of apoptosis was determined on the BD Accuri C6 flow cytometer (BD Biosciences) using the BD Accuri C6 Annexin V-FITC/PI template. Live, early, late apoptotic, as well as necrotic cells were quantified with the BD Accuri C6 software using the FITC signal detector FL1 (excitation = 488 nm; emission = 530 nm) for Annexin V positive cells and FL3 detector (excitation = 488 nm; emission = 670/LP) for PI positive cells.

### 4.8. Statistical Analysis

Data are represented as the mean ± standard error of the mean (SEM). Statistical comparisons between the control and different DMSO concentrations were performed using one-way analysis of variance (ANOVA), followed by a Tukey post hoc test and a student’s *t*-test using GraphPad Prism software version 5.0 (GraphPad Software, Inc., La Jolla, San Diego, CA, USA). Differences were regarded significant at *p* values < 0.05.

## 5. Conclusions

While DMSO remains an extensively used vehicle control and widespread solvent in numerous research settings, it is evident that its influence on cellular biological processes requires further investigation. Our findings demonstrate that DMSO doses higher than 0.001%, but not more than 0.5%, can still be safely used in experimental set-ups involving the use of H9c2 cells. However, these doses were determined to have some adverse effects in the breast cancer cells, as they stimulated the cells proliferative activity. Considering this, it may not be reasonable to recommend DMSO as a potential therapeutic compound for cancer cells. In the same context, it is important to note that when used as a solvent, the low DMSO doses may influence the study outcome, especially in experimental models where the proliferative activity of the cells is reduced by low FBS (2%) concentrations and when the treatment duration exceeds 6 days. Nonetheless, the observed effects of DMSO on the cardiac and cancer cells advocates for further investigation to fully understand the cytotoxic and beneficial effects of the “non-toxic” low DMSO doses.

## Figures and Tables

**Figure 1 molecules-26-07305-f001:**
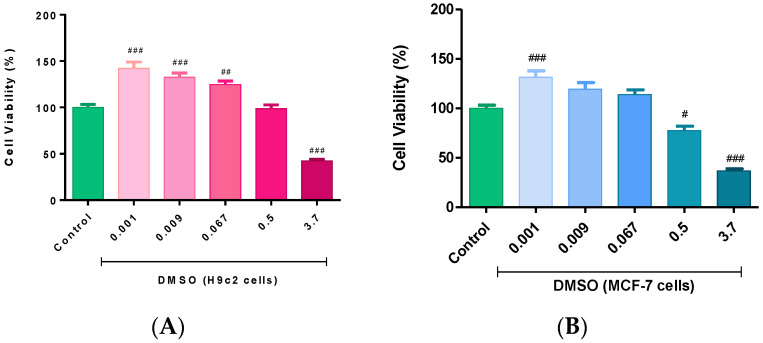
The effect of dimethyl sulfoxide (DMSO) on cell viability. An overview of the cytotoxicity effects of DMSO on the (**A**) H9c2 cardiomyoblasts and (**B**) MCF-7 breast cancer cells. Briefly, H9c2 and MCF-7 cells were treated with varying DMSO concentrations (0.001, 0.009, 0.067, 0.5, and 3.7%) every second day for 6 days. Untreated cells served as the control. Data are presented as the mean ± SEM of 3 biological experiments with 5 technical repeats (*n* = 3). Significance is indicated as ^#^
*p* < 0.05, ^##^
*p* < 0.01, and ^###^
*p* < 0.001 versus the control.

**Figure 2 molecules-26-07305-f002:**
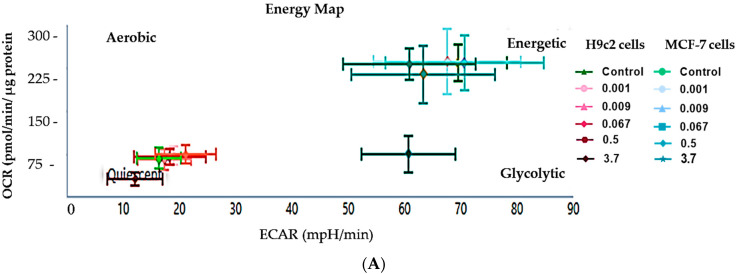
The effect of dimethyl sulfoxide (DMSO) on the mitochondrial bioenergetics in H9c2 cardiomyoblasts and MCF-7 breast cancer cells. (**A**) Energy phenotype, (**B**,**C**) Oxygen consumption rate (OCR), (**D**,**E**) extracellular acidification rate (**ECAR**), (**F**,**G**) basal respiration, (**H**,**I**) ATP-linked respiration, (**J**,**K**) ATP turnover, (**L**,**M**) maximal respiration and**,** (**N**,**O**) spare respiratory capacity**.** Both cell lines were treated every second day for 6 days with varying DMSO concentrations (0.001, 0.009, 0.067, 0.5, and 3.7%). Data are presented as the mean ± SEM of 3 biological experiments with 8 technical repeats (*n* = 3). Significance is indicated as ^#^
*p* < 0.05, ^##^
*p* < 0.01, and ^###^
*p* < 0.001 versus the control.

**Figure 3 molecules-26-07305-f003:**
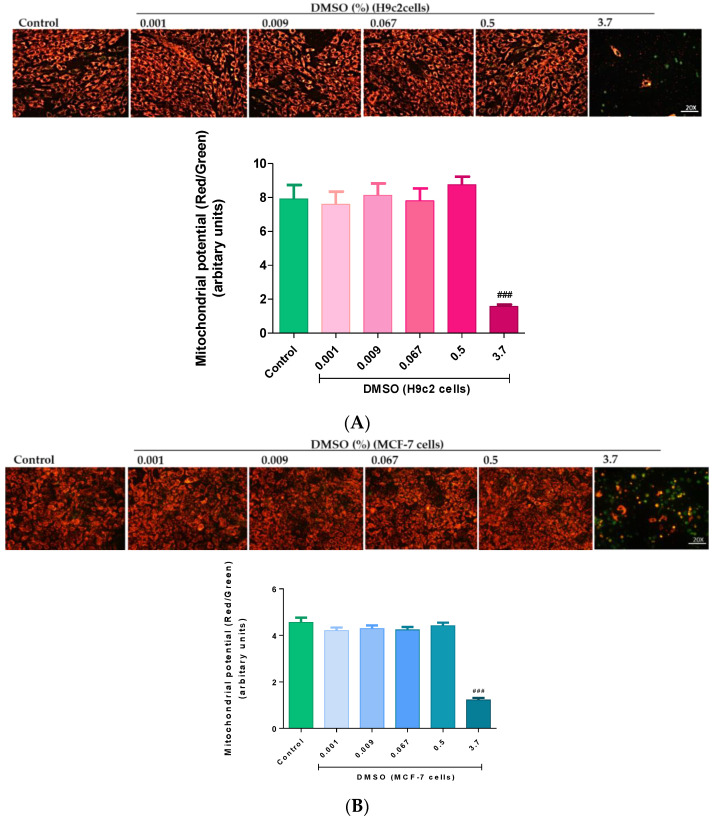
The effect of dimethyl sulfoxide (DMSO) the mitochondrial integrity of the (**A**) H9c2 cardiomyoblasts and (**B**) MCF-7 breast cancer cells. Cells were treated every second day for 6 days with varying DMSO concentrations (0.001, 0.009, 0.067, 0.5, and 3.7%). Data are presented as the mean ± SEM of 3 biological experiments with 5 technical repeats (*n* = 3). Significance is indicated as ^###^
*p* < 0.001 versus the control.

**Figure 4 molecules-26-07305-f004:**
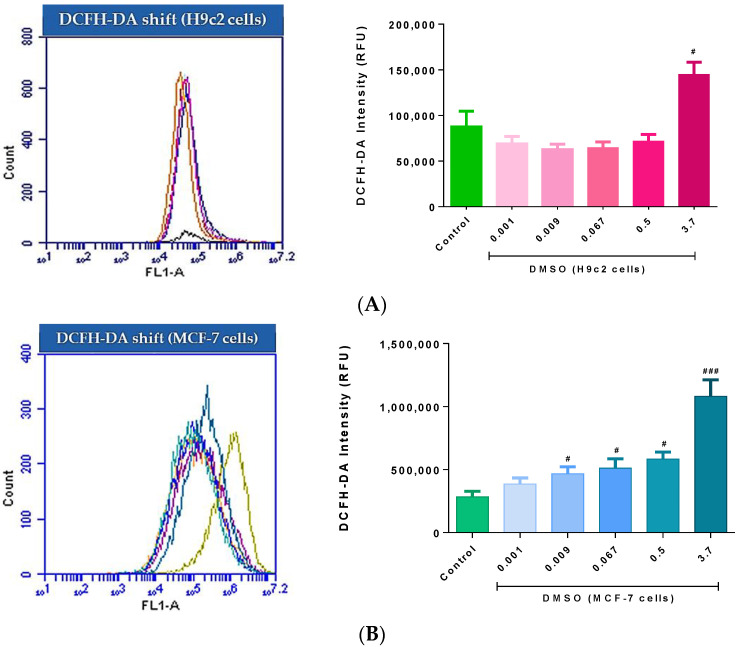
The effect of dimethyl sulfoxide (DMSO) on the production of reactive oxygen species (ROS) in (**A**) fluorescent shift of H9c2 cardiomyoblasts and (**B**) fluorescent shift of MCF-7 breast cancer cells. Cells were treated every second day for 6 days with varying DMSO concentrations (0.001, 0.009, 0.067, 0.5, and 3.7%). Data are presented as the mean ± SEM of 3 biological experiments with 3 technical repeats (*n* = 3). Significance is indicated as ^#^
*p* < 0.05, ^###^
*p* < 0.001 versus the control.

**Figure 5 molecules-26-07305-f005:**
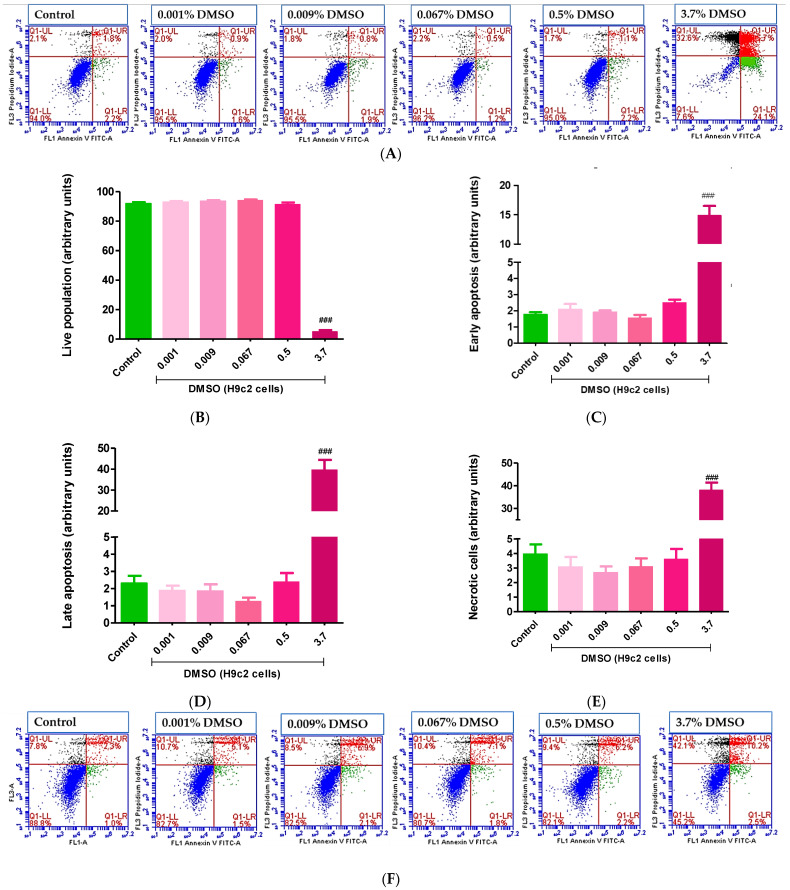
Dimethyl sulfoxide (DMSO)-induced apoptosis in the H9c2 cardiomyoblasts and MCF-7 breast cancer cells. (**A**) Flow cytometry scatter plot, (**B**) live population, (**C**) early apoptotic, (**D**) late apoptotic and (**E**) necrotic H9c2 cells. (**F**) Flow cytometry scatter plot, (**G**) live population, (**H**) early apoptotic, (**I**) late apoptotic and (**J**) necrotic MCF-7 cells. Cells were treated every second day for 6 days with varying DMSO concentrations (0.001, 0.009, 0.067, 0.5, and 3.7%). Data are presented as the mean ± SEM of 3 biological experiments with 3 technical repeats (*n* = 3). Data are presented as the mean ± SEM of 3 biological experiments with 3 technical repeats (*n* = 3). Significance is indicated as ^#^
*p* < 0.01, ^##^
*p* < 0.01, and ^###^
*p* < 0.001 versus the control.

**Figure 6 molecules-26-07305-f006:**
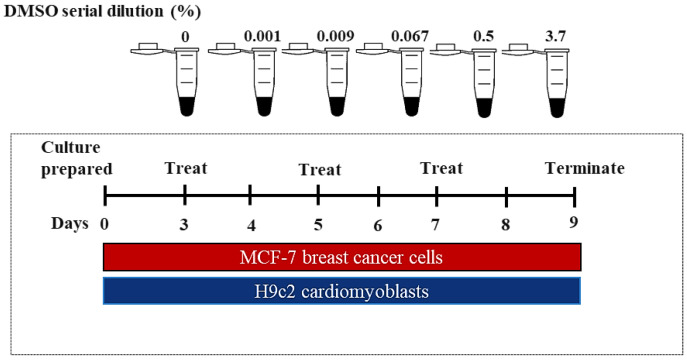
An in vitro experimental design of dimethyl sulfoxide (DMSO) exposure on H9c2 cardiomyoblasts and MCF-7 breast cancer cells. Concisely, cells were treated with varying DMSO concentrations (0.001 0.009, 0.067, 0.5, and 3.7%) for 6 days. Treatment was terminated on day 6 and biochemical analysis were conducted.

**Table 1 molecules-26-07305-t001:** Respiratory flux ratios of DMSO treated H9c2 cells.

Flux Ratios (pmol/min/µg Protein)	Treatment					
	Control	0.001	0.009	0.067	0.5	3.7
State _apparent_	3.6 ± 0.1	3.5 ± 0.1	3.5 ± 0.1	3.7 ± 0.2	3.4 ± 0.1	3.5 ± 0.1
Respiratory control ratio	6.96 ± 1.65	13.01 ± 0.79 ^#^	15.14 ± 1.84 ^##^	11.99 ± 1.24 ^#^	11.38 ± 0.81 ^#^	4,90 ± 1.20 ^#^
Coupling efficiency	0.91 ± 0.01	0.85 ± 0.04	0.79 ± 0.03	0.82 ± 0.01	0.83 ± 0.02	0.64 ± 0.03 ^###^

The state apparent, respiratory control ratio (RCR) and coupling efficiency of H9c2 cells in the presence or absence of DMSO were derived from the mitochondrial parameters presented in Figure 3A,B. Data represents mean ± SEM; *n* = 6. Significance is indicated as ^#^
*p* < 0.05, ^##^
*p* < 0.01, ^###^
*p* < 0.001 versus the control.

**Table 2 molecules-26-07305-t002:** Respiratory flux ratios of DMSO treated MCF-7 breast cancer cells.

Flux ratios (pmol/min/µg protein)	Treatment					
	Control	0.001	0.009	0.067	0.5	3.7
State _apparent_	3.58 ± 0.02	3.57 ± 0.02	3.57 ± 0.02	3.61 ± 0.01	3.65 ± 0.01	3.58 ± 0.08
Respiratory control ratio	13.95 ± 1.04	12.12 ± 0.59	12.14 ± 0.56	14.66 ± 0.55 ^#^	15.23 ± 0.97	9.26 ± 1.23 ^#^
Coupling efficiency	0.87 ± 0.01	0.85 ± 0.01	0.86 ± 0.01	0.83 ± 0.02	0.84 ± 0.02	0.79 ± 0.03 ^#^

The state apparent, respiratory control ratio (RCR) and coupling efficiency of MCF-7 cells in the presence or absence of DMSO were derived from the mitochondrial parameters presented in Figure 3A,B. Data represents mean ± SEM; *n* = 6. Significance is indicated as ^#^
*p* < 0.05 versus the control.

## Data Availability

Data is contained within the article or Appendix A The data presented in this study are available within the article and in the Appendix A. Design of the study; in the collection, analyses, or interpretation of data; in the writing of the manuscript, or in the decision to publish the results.

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
