# Peer review of "The Implication of Low Dose Dimethyl Sulfoxide on Mitochondrial Function and Oxidative Damage in Cultured Cardiac and Cancer Cells"

_molecules, 2021, doi:10.3390/molecules26237305_

Round 1
Reviewer 1 Report
The authors investigated effects of low doses of DMSO on cardiac and breast cancer cells. The study was focused on mitochondrial bioenergetics, oxidative stress, and apoptosis. The authors concluded that their results support the innocuous use of DMSO, as a potent solvent, for clinical research.
Although results are interesting and many parameters were determined, in my opinion the conclusions should be re-considered and re-written for MCF-7 cells. It is not reasonable to recommend DMSO as a potential therapeutic compound for cancer as it stimulates proliferation. MCF-7 cells treated with 0.001 (significantly), and 0.009 and 0.067% DMSO (non-significantly) proliferate better than control cells (Figure 1B, MTT assay), which is the undesirable effect for cancer cells. Hence, in the context of cancer therapy, usage of DMSO is questionable and this must be emphasized in the manuscript.
Line 17, Abstract – the authors wrote: “In the H9c2 cells, DMSO doses ≤ 0.5%enhanced mitochondrial bioenergetics and potential, whilst reducing ROS activity, early and late apoptosis.“ – in comparison with the control group this is not correct, misleading
Line 325 –There is also concern about seeding density. If H9c2 and MCF-t cells were analysed at day 9 after seeding, 1x105 is too high seeding density that may affect results (see https://www.thermofisher.com/hr/en/home/references/gibco-cell-culture-basics/cell-culture-protocols/cell-culture-useful-numbers.html). The same was evident for 96-well assays (line 290 -0.8 x 105/well in 96 well).
The introduction section must be improved as it does not provide information relevant for the study. Specific therapeutic benefits of DMSO, current uses and challenges must be better described, reasons for studying mitochondrial bioenergetics, previous findings related to mitochondrial function, oxidative stress and apoptosis, as well as the reason for choosing cardiomyoblasts and cancer cells.
Figure 3 – The collapse of the mitochondrial membrane potential is required for cytochrome c release. If mitochondrial membrane potential is preserved for MCF-7 cells at 0.009-0.5% DMSO, how they increase apoptotic rate?
Figure 4 – Is it possible to have the same RFU range for both cell lines? How is app. 600 counts represented as app. 85000 RFU for H9c2 cells, whereas app. 300 counts is represented as app. 250000 RFU for cancer cells?
In the Discussion section, some statements must be carefully checked (signs ≤ or < must be re-examined when indicating concentrations and their effects).
Regarding statistical analysis, there is no reason to compare effects of individual concentrations with 3.7% DMSO. All comparisons should be made with the control cells and discussed in that way.
There are also some minor inconsistencies.
Abstract, line 14 - DMSO doses should be kept below 0.001% - usually, 0.1% is considered as safe, please re-consider this value
Page 1, line 43 - Ref. 6 and 7 are not related to DMSO
Ref. 9 and lines 49-51 – Is it important to cite paper related to differentiation in the context of potential therapy?
Line 67 – the authors wrote that “The results obtained revealed a concentration-dependent cytotoxic effect of DMSO on both cell lines relative to the control group (p < 0.001).“ - for H9c2 cells cytotoxic effect was observed in only one dose (3.7%), it can not be concluded that the effect is concentration dependent
Line 94 – OCR is not represented on 2E. Capital letters should be put in the graphical representation.
Line 95 - ECAR is not represented on 2C
Figure 2 – all graphs (or fonts) are not of the same size
Line 96 - MCF-7 cells treated with DMSO doses ≤ 3.7% - should be at 3.7%?
Line 107 – The authors wrote - Surprisingly, we observed no significant changes in the MCF-7 cells respiratory flux ratios following DMSO exposure with doses ≤ 3.7% (Table 2). – levels of siginificance are indicated for 3.7%, unclear
Line 133 - These effects were comparable to the control groups of both H9c2 and 133 MCF-7 cells (Fig. 3A and B). – unclear
Line 167 - DMSO exposure with doses ≤ 3.7%....... – please check
Lines 111, 136, 154 – every
Line 275 – It is not clearly written if cells were under 2% FBS for 6 days. In that case, effects of low doses of FBS should also be considered.
Line 292 – 30 minutes for MTT? Usually cells are incubated for at least 3 hours
Method 4.5 – confusing; cells were seeded 0.8x105/well (what type of well?)
Lines 245-247: The authors wrote that „in the breast cancer cells, DMSO (≤ 3.7%) exposure impaired the cells oxidative phosphorylative capacity which was demonstrated by the cells reduced basal respiration, ATP-linked respiration, ATP turnover and maximal respiration.“ – this is not correct, the effect was observed only at 3.7%,
Line 342 – replace Sangweni et al. 2020 with the number
Author Response
Good day
Please see attachment.

Reviewer 2 Report
The manuscript submitted by Sangwani et,al. investigated the possible effect of low doses of DMSO on bioenergetics of two different cell lines representing cardiac and cancer cells. The study is interesting and well written. However, some concerns need to be clarified.
- The effect of DMSO was dose dependent, is there any justification for not performing a time dependent assessment to evaluate the possible effect at different time points, 24,48,72 h…etc (if they did, it worth including the Data). If not, their justification should be included in the discussion
- On what basis did the investigators choose 6 days of treatment? And their justification should be included in the discussion
- They conclude that when DMSO is used as a solvent, it may lead to false positive results, while DMSO when used as a solvent, usually the treatment does not last for six days, this conclusion should be revised.
- Results section 2.1.1. and figure 2 required an extensive revision, the labelling in the figure does not match the text in the results section (e.g. figure 2.e represent MCF7 while in the text line 94 it is used to refer to H9c2, same applies to the whole 2.1.1 section, this should be revised and correctly matched.
- Figure 2A, it was difficult to distinguish different line colors, please enhance the figure 2A
- In figure 2g, from the chart, it does not appear that at 0.001 and 0.5 there is any significance compared to control (##, and ###), same apply to figure 2i (at 0.5), 2.l (at 0.009) and 2.o (at 0.001), please revise and clarify?
- In Figure 2g and O, what significance is indicated by ##? (include in the legend)
- The observed cyto-toxic effect of 0.5 % of DMSO on MCF7 was not explained by any finding in this study, authors should discuss this in the discussion part.
- The discussion should be enhanced and revised to discuss all the findings in this study, for example, cyto-toxic finding, the different ORC results at 3.7 % (it represents a non-mitochondrial respiration) …etc.
Minor issues:
- The resolution of fig. 5 A and F should be improved.
- Fig 2 a,b,c..etc should be capitalised (Fig2.A, B, C…etc)
- In line 111 evert should be changed to every
- In line 342 the reference should be included correctly not as (Sangweni et al)
Author Response
Good day
Please see attachment.

Round 2
Reviewer 1 Report
Although the great amount of work has been done, and the authors tried to response to all queries, the fact is that they investigated the implication of low doses of dimethyl sulfoxide, but together with 2% FBS (because of the density). The usage of 2% FBS to hold proliferation is questionable with the original aim of the study. The results obtained are not comparable with other studies that usually use DMSO for a shorter period of time, and together with the 10% FBS. The study should be performed in a way to have low cell density in 10% FBS. Then it could be concluded if effects of DMSO were overlooked by other researchers.
Author Response
Although the great amount of work has been done, and the authors tried to response to all queries, the fact is that they investigated the implication of low doses of dimethyl sulfoxide, but together with 2% FBS (because of the density). The usage of 2% FBS to hold proliferation is questionable with the original aim of the study. The results obtained are not comparable with other studies that usually use DMSO for a shorter period of time, and together with the 10% FBS. The study should be performed in a way to have low cell density in 10% FBS. Then it could be concluded if effects of DMSO were overlooked by other researchers.
- Thank you for your comment. The authors acknowledge the reviewers' comment and have since included a cautionary statement on the use of low FBS content on how this may influence the experimental outcome. This has been highlighted in track changes in the conclusion section
Reviewer 2 Report
I would like to thank the authors for their responses. they addressed all my comments and suggestions,
Author Response
I would like to thank the authors for their responses. they addressed all my comments and suggestions,
- The authors would like to thank the reviewer for all the comments and suggestions.